# Socio-economic, demographic, and behavioural determinants of women's empowerment in Mozambique

Sofia Castro Lopes[1,2]*, Deborah Constant[1], Sílvia Fraga[3], Nafissa Bique Osman[4], Daniela Correia[3,5], Jane Harries[1]

1 Women's Health Research Unit, School of Public Health and Family Medicine, University of Cape Town, Cape Town, South Africa, 2 Division of Social and Behavioural Sciences, School of Public Health and Family Medicine, University of Cape Town, Cape Town, South Africa, 3 EPIUnit–Institute of Public Health, University of Porto, Porto, Portugal, 4 Department of Gynaecology and Obstetrics, Faculty of Medicine, Eduardo Mondlane University, Maputo, Mozambique, 5 Department of Public Health and Forensic Sciences, and Medical Education, Unit of Epidemiology, Faculty of Medicine, University of Porto, Porto, Portugal

* sofia.tclopes@gmail.com

## Abstract

### Introduction

Empowerment is considered pivotal for how women access and use health care services and experience their sexual and reproductive rights. In Mozambique, women's empowerment requires a better understanding and contextualization, including looking at factors that could drive empowerment in that context. This study aims to identify socioeconomic, demographic, and behavioural determinants of different domains of women's empowerment in Mozambique.

### Methods

Using the Demographic and Health Survey (DHS) conducted in 2015 for Mozambique, a sample of 2072 women aged between 15 and 49 years old were included in this study. The DHS's indicators of women's empowerment were used in a principal component analysis and the obtained components were identified as the domains of empowerment. Logistic regressions were run to estimate the association of socioeconomic, demographic, and behavioural characteristics with each domain of empowerment. Crude and adjusted odds ratios (OR) and respective 95% confidence intervals (95% CI) were calculated.

### Results

Three domains of women's empowerment were identified, namely (1) Beliefs about violence against women, (2) Decision-making, and (3) Control over sexuality and safe sex. Region, rurality, the experience of intimate partner violence (IPV) and partner's controlling behaviours were associated with Beliefs about violence against women, while Decision-making and Control over sexuality and safe sex were also associated with education, age and wealth. Employment, polygamous marriage and religion was positively associated with

**Data Availability Statement:** The data underlying the results presented in the study are available from the DHS Program website (https://

dhsprogram.com/data/Using-Datasets-for-Analysis.cfm).

**Funding:** We gratefully acknowledge the PhD grant from Fundação para a Ciência e a Tecnologia to SCL (SFRH/BD/146625/2019) and the contract to SF (CEECIND/01516/2017). We also thank the University of Cape Town, Faculty of Health Sciences for the Post Graduate Research Training Grant (FRC Award 2019) to SCL. The funders had no role in study design, data collection and analysis, decision to publish, or preparation of the manuscript.

**Competing interests:** The authors have declared that no competing interests exist.

Decision-making, and access to media increased the odds of Control over sexuality and safe sex.

## Conclusion

Women's empowerment seems to be determined by different socio-economic, demographic, and behavioural factors and this seems to be closely related to different domains of empowerment identified. This finding affirms the multi-dimensionality of empowerment as well as the importance of considering the context- and community-specific characteristics.

## Introduction

The Sustainable Development Goals (SDGs), launched in 2015, called for a global effort in the reduction of gender inequality and promotion of empowerment of women and girls, identifying these as priorities to be achieved in the next 15 years by countries around the world [1]. Women's empowerment is not only a mechanism to tackle gender inequalities, but also an end in itself by reinstating the opportunity for women to enjoy full sexual and reproductive health care and rights as well as have an active and recognized participation in society and the economy [1].

The definition of empowerment varies across the literature [2], but it can generally be described as the ability to exercise choice and free decision-making where this was previously denied [3]. Evidence suggests that empowered women are more able to make fertility decisions, use contraceptives and have increased communication with their partners [4, 5]. Empowerment results from the combination of two essential components, (1) preconditions such as education or income; and (2) agency, which consists of the actual act of choosing and making decisions. Education and income are described as essential preconditions to the process of empowerment [3, 6, 7], however, some studies suggest that there are other socioeconomic factors, namely women's age, age at marriage, income of the household, religion, access to land or property, among others, that can determine women's level of empowerment [8, 9].

Despite the recognition of the benefits of women's empowerment, the 2018 SDG report on goal number 5 showed that sociocultural norms and attitudes are persistent hindering factors in women's abilities to make free decisions [10]. At the basis of restrictive social and cultural norms and attitudes are gender power imbalances and inequalities, where women's decision-making regarding their own lives, health, reproduction and/or use of family planning is oftentimes undermined or non-existent [11, 12].

In Mozambique, the government is committed to tackling gender inequalities and power imbalances impacting on sexual and reproductive health and rights [13, 14], focussing among other aspects on addressing gender barriers to health care, increasing girls' access to education, as well as reduction of violence against girls. Despite these efforts, gender inequality is still a critical barrier and men remain the gatekeepers of decision-making related to women's sexual and reproductive health [11, 13, 15]. Women have little say and control about their fertility, the use of contraceptives and family planning [16]. Moreover, important indicators of women's health fall behind the international goals for Mozambique, including the maternal mortality ratio, estimated at 289 deaths per 100,000 live births in 2017 [17, 18], fertility levels (Fertility rate: 5.2 in 2016 and 4.8 in 2019) [17, 18] and prevalence of use of modern contraceptives (25% in 2015 to 35% in 2019) [19].

Empowerment is considered pivotal for how women access and use health care services and experience their sexual and reproductive rights [12]. Therefore, a better understanding of the process of women's empowerment and the factors associated with it could benefit health and gender strategies and interventions in Mozambique. This study aimed to identify the characteristics of women's life that can determine empowerment in Mozambique. To achieve this, the study was performed in two steps: (1) to identify the domains of women's empowerment in Mozambique and (2) to identify socioeconomic, demographic, and behavioural determinants of these domains of empowerment.

## Methods

### Data source

This study used data from the most recent Demographic Health Survey (DHS) conducted in Mozambique that included information needed for this analysis (2015). The DHS is part of a USAID program that supports countries to monitor and evaluate their demographic and health parameters at national and subnational levels [20].

The DHS 2015 for Mozambique was a population-based survey, including all 11 provinces which combined indicators from the HIV/AIDS indicator survey (AIS), the Malaria Indicator Survey (MIS) and the general DHS, including a range of indicators about population, health and nutrition [21]. Specifically, it encompassed information on socioeconomic and demographic characteristics of the participants, infant's vaccination, malaria, HIV (knowledge, testing and incidence and prevalence), fertility and fertility preferences, family planning, antenatal care, women's empowerment and domestic violence.

The DHS followed a rigorous population sampling process to ensure national, regional, urban, and rural representativeness. The survey was piloted in non-selected areas of the country, and changes made for improving clarity and adequacy of the questions. Interviewers received theoretical and practical training for the field work. A total of 25 teams were organised and distributed across the country. These included a supervisor, interviewers, and a person responsible to capture the data electronically.

Data was collected from all women and men, aged 15 to 59, residents or visitors that spent the night prior to the interview in one of the selected households. From the 7368 selected households, a total of 7129 were included in the survey, and 7749 women and 5283 men were interviewed [21]. Data were initially collected in paper forms, and immediately entered electronically in a data base.

This study includes women of reproductive age (15 to 49 years), who answered all sections of the survey, namely the section about empowerment including only women that were married or in a union, and the section on violence which was applied to a sub-sample of the female participants. Following these criteria, 2072 women were included in this analysis.

### Outcome variables

**Empowerment indicators.** The DHS 2015 survey was screened, and the relevant questions related to women's empowerment were identified. This process of identification was guided by current evidence available [8, 9, 22, 23], theoretical plausibility, and by the definition of empowerment used in this study. Empowerment was defined as having the power to control and freely decide over one's life and body to achieve valued or best-perceived outcomes. This definition is based on Kabeer's conceptualization of empowerment [3] and it incorporates the capability approach as a well-being measurement initially developed by Amartya Sen [24] and later adapted to health and empowerment studies [22].

The selected DHS empowerment indicators were related to decision-making within the household, justified beating and decisions about sexual intercourse. Similar to other approaches, these indicators were coded into a 3-point scale (i.e. values of -1, 0, 1) so that the highest value was given to categories considered to indicate greater level of empowerment [23]. This approach allowed to distinguish between women who were empowered in a particular area, from those who had some level of empowerment, and from those who were completely disempowered. More detail about the selected indicators of empowerment, and respective codes, used in the subsequent analysis can be found in the S1 Table.

### Independent variables

**Socioeconomic, demographic and behavioural indicators.** Using the same Mozambique DHS, socioeconomic, demographic and behavioural indicators that could be related to different empowerment levels were selected, guided by the WHO Social Determinants of Health Model, as well as recent evidence on determinants of women's empowerment [9, 11, 12, 25].

The socioeconomic and demographic variables included in the analysis were age (Less or equal to 19, 20–29, 30–39, 40–49 years), education (No education, Primary - 1st to 7th grade, secondary and above - 8th and above), current employment situation (Working, Not working), age of first co-habitation (10 to 14, 15 to 19, 20 and above years), polygamous marriage (Not polygamous, Polygamous, Doesn't know), religion (Catholic, Protestant, Islam, Evangelist, Zion, Other and non-religious).

The 11 provinces were used in the analysis and combined in three regions following the official aggregation of provinces by the Mozambique Government [26]: South–Maputo city, Maputo Province, Inhambane and Gaza; Centre–Sofala, Manica, Tete and Zambezia; and, North–Niassa, Cabo Delgado and Nampula. Urban or rural area of residency was also considered.

Wealth index, as computed by the DHS, is a composite measure based on the household cumulative living standards, namely the ownership of televisions and bicycles; the materials used for housing construction; and types of water access and sanitation facilities. Wealth index quintiles (poorest, poor, middle, rich, richest) was used in the analysis.

The indicator Access to media was created based on three variables which were the frequency (not at all, less than once a week, at least once a week) of reading a newspaper or magazine, of listening to the radio, and of watching TV. If the participant answered at least once a week to any of these options, it was coded 2, less than once a week it was coded 1, and if not at all it was coded as 0.

Women's exposure to controlling behaviours and domestic violence from their partners or husbands was also included in the analysis. The rationale for the inclusion of these behavioural indicators is the likelihood of these influencing the levels of empowerment of women, working most likely as a barrier to the process of empowerment. In some studies these indicators are included as measurements of empowerment [8, 27, 28], however, it does not fit the definition of empowerment used in this study. A variable showing the total number of controlling behaviours reported by women was computed and then transformed into a binary indicator, coded as "No control" and "At least one type of control". For domestic violence (hereafter: intimate partner violence), three variables were used to generate a new binary variable showing if women had ever experienced any type of violence perpetrated by her husband/partner.

### Data analysis

Descriptive measures of the socioeconomic, demographic and behavioural indicators and women's empowerment indicators by region were calculated, using cross-tabulation and chi-squared tests to compare proportions across regions.

Then, a principal component analysis (PCA) was carried out. PCA is a technique to transform a data set with a large number of indicators into a smaller data set of uncorrelated indicators, while capturing as much as possible of the variation of the original data set [29]. This procedure allows assessment of clustering patterns of empowerment indicators and the contribution (weight) for each component. PCA has been applied in studies on women's empowerment to avoid ad hoc estimation of summary scores in which each indicator has an equal contribution [8, 23, 30] From the scree plot of the PCA results, the significant components (eigenvalue above 1) were retained. An orthogonal varimax rotation was applied after confirming no correlation between the retained components, an essential criterium for this type of rotation [31–33]. The retained components represented the domains of women's empowerment identified for Mozambique. The Kaiser Meyer-Olkin (KMO) measure of sampling adequacy was then applied to test how suitable the data is for PCA.

Domain-specific empowerment indexes were calculated using the PCA factors scores. Each domain index was divided into quintiles from most empowered women (5th quintile) to least (1st quintile). The quintiles were then categorized as most vs the lesser empowered women (all groups below 5th quintile) for analysis [23]. Using logistic regression, we estimated the association of socioeconomic, demographic and behavioural characteristics and empowerment for each domain. The fit of the empowerment domains across regions was assessed through an interaction term/test between each domain and region. No significant differences were found therefore the results are presented together for all regions. The variable region was included in the final model.

Crude and adjusted odds ratios (OR) and respective 95% confidence intervals (95% CI) were calculated. STATA® version 16 [34] was used for all data analysis. The final models were adjusted for women's education, as research has shown that education is strongly associated with both empowerment and the other socio-economic, demographic, and behavioural characteristics included in the study. By adjusting for education, we aimed to assess if the associations found between the selected characteristics and empowerment were independent of the educational level of women. The inclusion of the different characteristics in the final models were informed by both theoretical and/or statistical justification (significance level set at 0.05).

## Results

The characteristics of the women included in the analysis are presented in Table 1. There were statistically significant differences among women from different regions across all socioeconomic, demographic and behavioural characteristics. Women from the South, including women from the capital city, Maputo, were slightly older, more educated, were likely to be employed, belonged to the richer or richest wealth quintiles and had more access to media than women from the centre and north regions. They were also older at the age of the first cohabitation and were less involved in polygamous marriages. On the other hand, women from the southern region were more exposed to controlling behaviours from the partner and intimate partner violence when compared to women from the centre and north regions (Table 1).

Table 2 describes the empowerment indicators of women at regional level, where some statistically significant differences could be observed. Generally, women from the south were more able to make decisions than those from the centre and north (p-value < .001) regions. Most women reported that beating is not justified in any situations across all regions. Interestingly, justified beating in any category was generally the highest in the south region; however, there were no statistically significant differences between regions, except in the category woman refuses to have sex (p-value = .005). Women from the south also reported more that

**Table 1. Socioeconomic, demographic and behavioural characteristics of women included in the study.**

| | Total | South | Centre | North |
|---|---|---|---|---|
| **N** | 2072 | 627 | 843 | 602 |
| **Age (years)** | | | | |
| Less or equal to 19 | 164 (7.9) | 25 (4.0) | 74 (8.8) | 65 (10.8) |
| 20–29 | 849 (41.0) | 241 (38.4) | 372 (44.1) | 236 (39.2) |
| 30–39 | 626 (30.2) | 220 (35.1) | 245 (29.1) | 161 (26.7) |
| 40–49 | 433 (20.9) | 141 (22.5) | 152 (18.0) | 140 (23.3) |
| **Education[1]** | | | | |
| No education | 604 (29.2) | 106 (16.9) | 292 (34.6) | 206 (34.2) |
| Primary (1st to 7th grade) | 1086 (52.4) | 374 (59.7) | 410 (48.6) | 302 (50.2) |
| Secondary and above (8th and above) | 382 (18.4) | 147 (23.4) | 141 (16.7) | 94 (15.6) |
| **Currently employed** | | | | |
| Yes | 912 (44.0) | 334 (53.3) | 361 (42.8) | 217 (36.1) |
| **Age of first cohabitation[2] (years)** | | | | |
| 10 to 14 | 368 (17.8) | 50 (8.0) | 152 (18.0) | 166 (27.6) |
| 15 to 19 | 1154 (55.7) | 363 (57.9) | 470 (55.8) | 321 (53.3) |
| 20 or above | 550 (26.5) | 214 (34.1) | 221 (26.2) | 115 (19.1) |
| **Polygamous marriage** | | | | |
| No polygamous | 1603 (77.4) | 515 (82.1) | 610 (72.4) | 478 (79.4) |
| Polygamous | 406 (19.6) | 81 (12.9) | 206 (24.4) | 119 (19.8) |
| Does not know | 63 (3.0) | 31 (4.9) | 27 (3.2) | 5 (0.8) |
| **Urban vs rural residency** | | | | |
| Rural | 1321 (63.8) | 338 (53.9) | 574 (68.1) | 409 (67.9) |
| **Religion** | | | | |
| Catholic | 494 (23.8) | 94 (15.0) | 161 (19.1) | 239 (39.7) |
| Protestant | 402 (19.4) | 169 (27.0) | 219 (26.0) | 14 (2.3) |
| Islamic | 372 (18.0) | 19 (3.0) | 39 (4.6) | 314 (52.6) |
| Evangelical | 254 (12.3) | 133 (21.2) | 105 (12.5) | 16 (2.7) |
| Zion | 299 (14.4) | 152 (24.2) | 145 (17.2) | 2 (0.3) |
| Other | 75 (3.6) | 17 (2.7) | 48 (5.7) | 10 (1.7) |
| No-religion | 176 (8.5) | 43 (6.9) | 126 (15.0) | 7 (1.2) |
| **Wealth index** | | | | |
| Poorest | 337 (16.3) | 12 (1.9) | 162 (19.2) | 163 (27.1) |
| Poorer | 381 (18.4) | 22 (3.5) | 204 (24.2) | 155 (25.8) |
| Middle | 430 (20.8) | 105 (16.8) | 204 (24.2) | 121 (20.1) |
| Rich | 475 (22.9) | 233 (37.2) | 143 (17.0) | 99 (16.5) |
| Richest | 449 (21.7) | 255 (40.7) | 130 (15.4) | 64 (10.6) |
| **Access to media[3]** | | | | |
| No access | 978 (47.3) | 226 (36.0) | 406 (48.2) | 346 (57.7) |
| Less than once a week | 366 (17.7) | 86 (13.7) | 180 (21.4) | 100 (16.7) |
| At least once a week | 726 (35.1) | 315 (50.2) | 257 (30.5) | 154 (25.7) |
| **Partner controlling behavior** | | | | |
| At least one type | 881 (42.5) | 311 (49.6) | 330 (39.2) | 240 (39.9) |
| **IPV* exposure[4]** | | | | |

*(Continued)*

**Table 1.** (Continued)

|  | Total | South | Centre | North |
|---|---|---|---|---|
| **N** | **2072** | **627** | **843** | **602** |
| Yes | 471 (22.8) | 181 (29.1) | 189 (22.4) | 101 (16.8) |

Note: All subcategories for each variable are statistically significant at a p-value .001.

[1] Based on the previous education system organization. System changed in 2018.

[2] Included only married or ever married women.

[3] Missing = 2

[4] Missing = 5

*IPV–Intimate partner violence.

women can ask the partner for the use of a condom in sexual intercourse and refuse sex, while the centre region presented the lowest percentage (p-value < .001).

With the PCA three significant components were retained. The three retained components explained 25%, 19%, and 16% of the total variance, respectively, adding up to 60%. The KMO test value was 0.75 therefore we consider the sampling adequate for PCA. The retained components were then identified as empowerment domains and included: Beliefs about violence against women; Decision-making; and Control over sexuality and safe sex. The factor loadings of each indicator within each component are presented on S2 Table.

Table 3 shows the crude and adjusted OR for the association between the socio-economic, demographic, and behavioural characteristics and the different domains of empowerment.

After adjusting for woman's education, we observed that age, education, current employment, age of first cohabitation, polygamous marriage and the wealth index were not associated with the domain Beliefs about violence against women. However, experiencing at least one type of controlling behaviour, being exposed to IPV, having access to media, and living in the South region of Mozambique was significant and negatively associated with being empowered in this domain, which seems to indicate that these factors are determinants of lower levels of empowerment for Beliefs about violence against women. Rurality had significant and positive impact on this domain.

After adjusting for education, Decision-making domain of empowerment was significantly and positively associated with women of older age, more educated, currently working, living in South or Centre regions and with increased levels of wealth (Table 3). IPV and controlling behaviours from the partners, were also statistically and positively associated with higher decision-making power. No associations were found between this domain of empowerment and age of first cohabitation, access to media and rural vs. urban residency.

Current employment, age at first cohabitation, polygamous marriage and religion were not associated with Control over Sexuality and safe sex after adjusting for women's education. However, having some education, living in the South region, being among the richest wealth quintile, having access to media at least once a week as well as experiencing IPV or partner's controlling behaviour had a significant and positive impact in women's empowerment level for this domain, after adjusting for education (Table 3). Being 40 to 49 years old and living in a rural area were significantly and negatively associated with women's control over their sexuality.

## Discussion

This study identified three domains of women's empowerment for Mozambique which included Beliefs about violence against women, Decision-making and Control over sexuality

**Table 2. Indicators of empowerment among women by region.**

| Indicators | Total | South | Centre | North | p-value |
|---|---|---|---|---|---|
| N | 2072 | 627 | 843 | 602 | |
| **Who usually decides on:** | | | | | |
| **woman's health care** | | | | | |
| Woman alone | 428 (20.7) | 165 (26.3) | 189 (22.4) | 74 (12.3) | < .001 |
| Jointly | 1271 (61.3) | 396 (63.2) | 474 (56.2) | 401 (66.6) | |
| Partner or other alone | 373 (18.0) | 66 (10.5) | 180 (21.4) | 127 (21.1) | |
| **large purchases for the household** | | | | | |
| Woman alone | 461 (22.3) | 184 (29.4) | 207 (24.6) | 70 (11.6) | < .001 |
| Jointly | 1168 (56.4) | 373 (59.5) | 429 (50.9) | 366 (60.8) | |
| Partner or other alone | 443 (21.4) | 70 (11.2) | 207 (24.6) | 166 (27.6) | |
| **visit family and friends** | | | | | |
| Woman alone | 396 (19.1) | 135 (21.5) | 164 (19.5) | 97 (16.1) | < .001 |
| Jointly | 1297 (62.6) | 407 (64.9) | 488 (57.9) | 402 (66.8) | |
| Partner or other alone | 379 (18.3) | 85 (13.6) | 191 (22.7) | 103 (17.1) | |
| **Beating justified if:** | | | | | |
| **wife goes out without telling husband** | | | | | |
| Not Justified | 1890 (91.2) | 576 (91.9) | 755 (89.6) | 559 (92.9) | 0.070 |
| Don't know | 13 (0.6) | 1 (0.2) | 9 (1.1) | 3 (0.5) | |
| Justified | 169 (8.2) | 50 (8.0) | 79 (9.4) | 40 (6.6) | |
| **wife neglects the children** | | | | | |
| Not Justified | 1976 (95.4) | 597 (95.2) | 811 (96.2) | 568 (94.4) | 0.174 |
| Don't know | 12 (0.6) | 2 (0.3) | 7 (0.8) | 3 (0.5) | |
| Justified | 84 (4.1) | 28 (4.5) | 25 (3.0) | 31 (5.2) | |
| **wife argues with husband** | | | | | |
| Not Justified | 1924 (92.9) | 581 (92.7) | 783 (92.9) | 560 (93.0) | 0.172 |
| Don't know | 18 (0.9) | 1 (0.2) | 10 (1.2) | 7 (1.2) | |
| Justified | 130 (6.3) | 45 (7.1) | 50 (5.9) | 35 (5.8) | |
| **wife refuses to have sex** | | | | | |
| Not Justified | 1912 (92.3) | 568 (90.6) | 795 (94.3) | 549 (91.2) | 0.005 |
| Don't know | 32 (1.5) | 6 (1.0) | 14 (1.7) | 12 (2.0) | |
| Justified | 128 (6.2) | 53 (8.5) | 34 (4.0) | 41 (6.8) | |
| **wife burns the food** | | | | | |
| Not Justified | 2008 (96.9) | 609 (97.1) | 823 (97.6) | 576 (95.7) | 0.070 |
| Don't know | 19 (0.9) | 2 (0.3) | 8 (1.0) | 9 (1.5) | |
| Justified | 45 (2.2) | 16 (2.6) | 12 (1.4) | 12 (2.8) | |
| **A woman can:** | | | | | |
| **ask husband/partner to use condom if he has STI[1]** | | | | | |
| No | 504 (24.3) | 98 (15.6) | 267 (31.7) | 139 (23.1) | < .001 |
| Yes | 1277 (61.6) | 498 (79.4) | 430 (51.0) | 349 (58.0) | |
| Does not know | 291 (14.0) | 31 (4.9) | 146 (17.3) | 114 (18.9) | |
| **ask husband/partner to use condom** | | | | | |
| No | 737 (35.7) | 140 (22.3) | 380 (45.1) | 217 (36.1) | < .001 |
| Yes | 1090 (52.6) | 449 (71.6) | 342 (40.6) | 299 (49.7) | |
| Does not know | 245 (11.8) | 38 (6.1) | 121 (14.4) | 86 (14.3) | |
| **refuse sex** | | | | | |
| No | 547 (26.4) | 104 (16.6) | 313 (37.1) | 130 (21.6) | < .001 |
| Yes | 1378 (66.5) | 504 (80.4) | 433 (51.4) | 441 (73.3) | |

(*Continued*)

**Table 2.** (Continued)

| Indicators | Total | South | Centre | North | p-value |
|---|---|---|---|---|---|
| **N** | **2072** | **627** | **843** | **602** | |
| Does not know | 147 (7.1) | 19 (3.0) | 97 (11.5) | 31 (5.2) | |

[1] STI, Sexually transmitted infection.

and safe sex. Similar results were found in other studies conducted in African contexts, despite differences in the order of and contribution of each indicator for each domain [23, 35, 36]. Despite the benefits of having standardized and comparable data across countries, DHS has few empowerment indicators, which may limit its capacity to grasp contextual specificities [4, 23]. Some studies attempt to overcome this limitation by using other indicators as proxies of empowerment, such as women's education, income or access to information. However, this has been identified as problematic, raising issues around conceptualization and operationalization of empowerment and contributing to inconclusive results [2, 4, 5, 35]. In order to minimize this limitation, a conservative approach was adopted in this study, where the selected indicators of women's empowerment included only those linked to actions or beliefs which could lead to actions taken by women themselves, and then we identified associated factors that may contribute to improving each empowerment domain of women's lives.

Our findings suggest that the domain of empowerment Beliefs about violence against women is shaped by the community and contextual determinants like region, place of residency (rural vs. urban) and the partner's behaviour, rather than by women's individual characteristics. The importance of community factors for impeding or facilitating women's empowerment and its relationship with violence against women has been described in other contexts [37]. Additional to community and/or contextual determinants, individual characteristics of women like education, age or wealth, also seem to play a role in determining empowerment in the domain Decision-making and Control over sexuality and safe sex in Mozambique, similar to findings from other studies [35, 36]. Our findings could be explained by the fact that women's individual beliefs are rooted in socio-cultural norms and traditional practices embedded in patriarchal systems, learned and maintained by the community where women live [38, 39] and oftentimes perpetuated by women themselves [37, 39, 40]. Despite the matrilineal societal organization of the north region, Mozambique is a patriarchal society with rigid gender norms that retain men in power positions [41, 42]. While education, wealth, age, and employment are assets or resources that women use in the process of decision-making and choice [36], therefore playing an important role in the other two domains: Decision-making and Control over sexuality and safe sex.

Partners controlling behaviour and IPV were found to be important determinants of curtailing women's empowerment in Mozambique. This is aligned with findings from studies in other sub-Saharan African countries [38, 43–45]. A study involving 17 African sub-Saharan countries, including Mozambique, showed that IPV is socially and culturally acceptable, giving the partner the right to control and "correct" an erring wife or woman [39]. However, when women are empowered for Decision-making and Control over sexuality, they likely become enabled to identify and report on abusive experiences [38]. Furthermore, available evidence suggests that when women enter the pathway of empowerment they may challenge gender norms and gender power relations, which might initially expose them to a greater risk of experiencing violence and controlling behaviours perpetrated by partners, referred to as violence backlash [37, 38, 43, 46]. However, there is evidence suggesting that empowerment can become protective against IPV throughout time, where the empowered women are less likely

**Table 3. Determinants of most empowered women for each domain of empowerment.**

| | Beliefs about violence against women | | Decision-making | | Control over sexuality and sex | |
|---|---|---|---|---|---|---|
| | cOR (95% CI) | aOR (95% CI) [1] | cOR (95% CI) | aOR (95% CI) [1] | cOR (95% CI) | aOR (95% CI) [1] |
| **Age (years)** | | | | | | |
| Less or equal to 19 | 1 | 1 | 1 | 1 | 1 | 1 |
| 20–29 | 0.89 (0.60, 1.33) | 0.90 (0.61, 1.35) | **1.59 (1.00, 2.53)** | 1.56 (0.98, 2.48) | 0.81 (0.51, 1.28) | 0.78 (0.49, 1.24) |
| 30–39 | 0.68 (0.44, 1.03) | 0.70 (0.46, 1.06) | **1.82 (1.13, 2.92)** | **1.96 (1.22, 3.16)** | 0.64 (0.39.1.03) | 0.72 (0.44, 1.17) |
| 40–49 | 0.76 (0.49, 1.18) | 0.78 (0.50, 1.21) | 1.62 (0.99, 2.64) | **1.76 (1.10, 2.90)** | **0.45 (0.26, 0.76)** | **0.51 (0.30, 0.89)** |
| **Education** | | | | | | |
| No education | 1 | 1 | 1 | 1 | 1 | 1 |
| Primary | 1.24 (0.96, 1.61) | 1.24 (0.96, 1.61) | 1.21 (0.94, 1.56) | 1.21 (0.94, 1.56) | **1.69 (1.18, 2.40)** | **1.69 (1.18, 2.41)** |
| Secondary or above | 1.16 (0.84, 1.61) | 1.16 (0.84, 1.61) | **1.68 (1.24, 2.28)** | **1.68 (1.24, 2.28)** | **2.98 (2.01, 4.43)** | **2.98 (2.01, 4.43)** |
| **Currently employed** | | | | | | |
| No | 1 | 1 | 1 | 1 | 1 | 1 |
| Yes | 0.97 (0.78, 1.21) | 0.97 (0.78, 1.21) | **1.64 (1.33, 2.03)** | **1.65 (1.33, 2.03)** | 0.94 (0.72, 1.22) | 0.93 (0.71, 1.22) |
| **Age of first cohabitation (years)** | | | | | | |
| 10 to 14 | 1 | 1 | 1 | 1 | 1 | 1 |
| 15 to 19 | 1.05 (0.78, 1.42) | 1.03 (0.76, 1.38) | 0.81 (0.62, 1.08) | 0.76 (0.58, 1.01) | 1.39 (0.94, 2.06) | 1.22 (0.82, 1.83) |
| 20 or above | 0.95 (0.68, 1.33) | 0.94 (0.67, 1.33) | 0.89 (0.65, 1.22) | 0.82 (0.60, 1.12) | 1.48 (0.96, 2.28) | 1.26 (0.81, 1.95) |
| **Polygamous marriage** | | | | | | |
| No polygamous | 1 | 1 | 1 | 1 | 1 | 1 |
| Polygamous | 0.91 (0.69, 1.20) | 0.92 (0.70, 1.22) | **1.60 (1.25, 2.06)** | **1.73 (1.34, 2.24)** | **0.61 (0.42, 0.90)** | 0.69 (0.47, 1.02) |
| Does not know | 0.76 (0.38, 1.50) | 0.77 (0.39, 1.52) | **2.73 (1.63, 4.60)** | **2.80 (1.66, 4.73)** | 0.84 (0.38, 1.86) | 0.86 (0.38, 1.92) |
| **Region** | | | | | | |
| North | 1 | 1 | 1 | 1 | 1 | 1 |
| Centre | 0.88 (0.68, 1.14) | 0.88 (0.68, 1.15) | **1.63 (1.23, 2.16)** | **1.62 (1.22, 2.16)** | 1.17 (0.82, 1.67) | 1.17 (0.82, 1.67) |
| South | 0.77 (0.58, 1.03) | **0.74 (0.55, 0.99)** | **2.56 (1.92, 3.42)** | **2.47 (1.85, 3.30)** | **2.00 (1.41, 2.83)** | **1.80 (1.26, 2.56)** |
| **Rural vs Urban** | | | | | | |
| Urban | 1 | 1 | 1 | 1 | 1 | 1 |
| Rural | 1.24 (0.98, 1.56) | **1.36 (1.05, 1.75)** | 0.85 (0.69, 1.10) | 1.00 (0.79, 1.28) | **0.51 (0.39, 0.66)** | **0.65 (0.48, 0.87)** |
| **Religion** | | | | | | |
| Catholic | 1 | 1 | 1 | 1 | 1 | 1 |
| Islamic | 0.92 (0.66, 1.28) | 0.92 (0.66, 1.29) | 0.78 (0.54, 1.12) | 0.80 (0.55, 1.15) | 0.70 (0.45, 1.09) | 0.73 (0.47, 1.14) |
| Protestant | 0.86 (0.62, 1.20) | 0.87 (0.62, 1.20) | **1.51 (1.10, 2.08)** | **1.54 (1.12, 2.13)** | 1.17 (0.80, 1.73) | 1.23 (0.83, 1.81) |
| Evangelical | 0.84 (0.57, 1.23) | 0.83 (0.56, 1.21) | **1.71 (1.19, 2.44)** | **1.67 (1.17, 2.40)** | 1.00 (0.64, 1.59) | 0.95 (0.60, 1.51) |
| Zion | 0.95 (0.67, 1.35) | 0.95 (0.66, 1.35) | **1.56 (1.10, 2.20)** | **1.67 (1.17, 2.37)** | 0.92 (0.59, 1.44) | 1.04 (0.67, 1.63) |
| Other | **0.38 (0.17, 0.85)** | **0.38 (0.17, 0.86)** | 1.42 (0.80, 2.52) | 1.48 (0.83, 2.64) | 1.33 (0.68, 2.60) | 1.44 (0.73, 2.84) |
| No religion | 0.92 (0.60, 1.41) | 0.94 (0.61, 1.45) | 1.24 (0.81, 1.89) | 1.35 (0.88, 2.08) | 0.70 (0.39, 1.24) | 0.83 (0.46, 1.50) |
| **Wealth index** | | | | | | |
| Poorest | 1 | 1 | 1 | 1 | 1 | 1 |
| Poorer | 1.13 (0.79, 1.62) | 1.10 (0.77, 1.58) | 1.47 (1.00, 2.16) | 1.45 (0.98, 2.14) | 0.88 (0.50, 1.54) | 0.85 (0.48, 1.50) |
| Middle | 0.96 (0.67, 1.37) | 0.92 (0.65, 1.32) | **1.50 (1.03, 2.20)** | **1.47 (1.01, 2.16)** | 1.29 (0.78, 2.16) | 1.24 (0.74, 2.08) |
| Richer | 0.81 (0.57, 1.56) | 0.73 (0.51, 1.06) | **1.65 (1.14, 2.38)** | **1.53 (1.05, 2.23)** | **1.73 (1.07, 2.80)** | 1.51 (0.92, 2.48) |
| Richest | 0.88 (0.62, 1.26) | 0.75 (0.50, 1.12) | **2.05 (1.42, 2.94)** | **1.75 (1.17, 2.62)** | **3.21 (2.02, 5.08)** | **2.54 (1.53, 4.21)** |
| **Access to media** | | | | | | |
| No access | 1 | 1 | 1 | 1 | 1 | 1 |
| Less than once a week | 0.82 (0.61, 1.11) | 0.79 (0.59, 1.08) | 1.28 (0.96, 1.71) | 1.23 (0.93, 1.65) | 1.36 (0.93, 2.01) | 1.27 (0.86, 1.88) |
| At least once a week | **0.75 (0.59, 0.96)** | **0.68 (0.52, 0.89)** | **1.27 (1.01, 1.60)** | 1.09 (0.84, 1.41) | **2.01 (1.49, 2.70)** | **1.56 (1.13, 2.15)** |
| **Partner controlling behavior** | | | | | | |

*(Continued)*

**Table 3.** (Continued)

| | Beliefs about violence against women | | Decision-making | | Control over sexuality and sex | |
|---|---|---|---|---|---|---|
| | cOR (95% CI) | aOR (95% CI) [1] | cOR (95% CI) | aOR (95% CI) [1] | cOR (95% CI) | aOR (95% CI) [1] |
| No control | 1 | 1 | 1 | 1 | 1 | 1 |
| At least one type | 0.70 (0.56, 0.87) | 0.69 (0.55, 0.87) | 1.45 (1.17, 1.79) | 1.41 (1.14, 1.74) | 2.10 (1.60, 2.75) | 1.99 (1.52, 2.61) |
| **IPV exposure** | | | | | | |
| No | 1 | 1 | 1 | 1 | 1 | 1 |
| Yes | 0.72 (0.54, 0.95) | 0.71 (0.54, 0.93) | 1.64 (1.30, 2.08) | 1.58 (1.25, 2.01) | 1.51 (1.13, 2.03) | 1.39 (1.03, 1.87) |

[1]Adjusted for education.

to be perceived as transgressing the gender norms [47]. Two conclusions can be drawn from these results: each domain of empowerment is measuring different aspects of a woman's life, and there are unique pathways towards empowerment for each domain.

Education and employment have been consistently described in the literature as a key element for women's empowerment [48]. Some studies have shown that the effect of women's education on institutional delivery was mediated by the different domains of empowerment [35, 36]. Our results are consistent with the literature about the role of education, however, they showed that being employed was only associated with Decision-making. The available evidence for the African context raises questions about the contribution of employment for women's status and empowerment [35, 36]. In some sub-Saharan African countries, where women get paid not only in cash but also in-kind, or not paid at all, it is possible that not all of these offer a way to empowerment. Furthermore, women may not have the power to manage the generated income. Wealth and age were also associated with women's Decision-making and Control over sexuality and safe sex domains of empowerment, similar to what has been described in the literature for different women's empowerment domains [35, 49, 50].

Our findings suggest that the region plays a role in determining the level of empowerment in different domains while the role of place of residency (rural vs urban) is less clear. These findings are aligned with the available evidence [9, 35, 36, 43, 44]. The negative association between tolerance to violence against women and the south region could be evidence of the patrilineal organization of the southern region of Mozambique [42]. However, the higher access to education and urbanization of the south could promote more positive gender-based views when comparing to the north and centre regions. Despite the observed differences across regions, in the final model, the variable region did not change the associations found (results not showed). Notwithstanding, the region could be reflecting cultural specificities that should be considered when putting forward interventions aiming to empowerment women.

The current evidence for the relationship between access to media and empowerment is inconclusive with studies reporting different or no association [12, 39, 45]. The media are both the vehicle of messages promoting gender equality and preventing violence against women as well as a place that perpetuates messages based on gender norms and gender inequalities based on the culture and social beliefs [44, 51], and this in part explains the positive and negative relationships with empowerment found in our study. There is a need for further research and analysis on media and women's empowerment in Mozambique.

## Strengths and limitations

The study's strength relates to the use of a large sample of women of reproductive age, from a population-based survey, which allowed generalizability of the findings for Mozambique.

Nevertheless, the study has some limitations that should be considered. First, the DHS is a cross-sectional survey hence it cannot be used to infer causality. As empowerment is the process of gaining power, its effect should ideally be examined using longitudinal data which would allow perspective over time. Second, empowerment indicators available in the DHS involved only partnered women. This is related to a gap in empowerment literature about young and/or unmarried women. Empowered women might not be married or may marry later in life, but there is a persistent lack of data and specific indicators targeting this group of women [23]. A third limitation was the use of a cut-off point in the analysis to identify and define women with a high level of empowerment. There is no evidence available supporting how and what level of women's empowerment should be considered high or satisfactory, however, the use of quintiles offered a consistent way of doing that (the fifth quintile implying empowered women), and has been used in previous studies [23]. Finally, this study focused on quantitative measures of empowerment, limited by the available data. It is possible that this data is not capturing all the domains of empowerment in the context of Mozambique. Further research, in particular qualitative research, should be conducted to fully explore domains and determinants of empowerment.

## Conclusion

In Mozambique, women's empowerment seems to be determined by socio-economic, demographic, and behavioural factors, and this seems to be closely related to the different domains of empowerment identified. This finding affirms the multidimensionality of empowerment as well as the importance of considering context- and community-specific characteristics. Education of women and girls seems to play an important role for empowerment, and is an area requiring continued investment not only by the Mozambican government but also by organisations working in women's empowerment. Not surprisingly, IPV and partners' controlling behaviours were found to be important barriers to women's empowerment. The region of residency due to cultural and societal organization differences was shown to play a crucial role in women's empowerment. This not only highlights the need to capture the nuances of empowerment in each context, but also for the need to tailor and contextualise interventions and programmes. This study offers a step forward in understanding women's empowerment in Mozambique, however further studies are needed, particularly of a qualitative nature, to explore women's understanding and experiences of empowerment.

## Supporting information

**S1 Table. Indicators of empowerment identified in Mozambique DHS 2015.**
(DOCX)

**S2 Table. Retained components' factor loadings after orthogonal varimax rotation.**
(DOCX)

## Acknowledgments

We are grateful to the Demographic and Health Survey Program for sharing and authorizing the use of the 2015 dataset for Mozambique for this study.

## Author Contributions

**Conceptualization:** Sofia Castro Lopes, Deborah Constant, Sílvia Fraga, Jane Harries.

**Formal analysis:** Sofia Castro Lopes, Deborah Constant, Daniela Correia.

**Funding acquisition:** Sofia Castro Lopes, Deborah Constant, Sílvia Fraga, Jane Harries.

**Methodology:** Sofia Castro Lopes, Deborah Constant, Sílvia Fraga, Jane Harries.

**Writing – original draft:** Sofia Castro Lopes, Deborah Constant, Sílvia Fraga, Nafissa Bique Osman, Daniela Correia, Jane Harries.

**Writing – review & editing:** Sofia Castro Lopes, Deborah Constant, Sílvia Fraga, Nafissa Bique Osman, Daniela Correia, Jane Harries.

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
