## [Decision Letter · Decision Letter 0]

24 Nov 2020

PONE-D-20-25980

Socio-economic, demographic, and behavioural determinants of women’s empowerment in Mozambique.

PLOS ONE

Dear Dr. Sofia Castro Lopes

Thank you for submitting your manuscript to PLOS ONE. After careful consideration, we feel that it has merit but does not fully meet PLOS ONE’s publication criteria as it currently stands. Therefore, we invite you to submit a revised version of the manuscript that addresses the points raised during the review process.

Please take cognisance of the comments of Reviewer 2 and rework your manuscript.

We look forward to receiving your revised manuscript.

Kind regards,

Eugene Kofuor Maafo Darteh, Ph.D.

Academic Editor

PLOS ONE

Journal Requirements:

Reviewers' comments:

Reviewer's Responses to Questions

**Comments to the Author**

1. Is the manuscript technically sound, and do the data support the conclusions?

Reviewer #1: Yes

Reviewer #2: No

2. Has the statistical analysis been performed appropriately and rigorously? 

Reviewer #1: Yes

Reviewer #2: Yes

3. Have the authors made all data underlying the findings in their manuscript fully available?

Reviewer #1: Yes

Reviewer #2: Yes

4. Is the manuscript presented in an intelligible fashion and written in standard English?

Reviewer #1: Yes

Reviewer #2: Yes

5. Review Comments to the Author

Reviewer #1: Overall this is an important topic and a well constructed manuscript that goes beyond limited linear notions of empowerment. Please see Gram et al., 2018 https://doi.org/10.1007/s11205-018-2012-2

"Organising concepts of ‘women’s empowerment for measurement : a typology", which will help lay out the foundation for your introduction. See minor comments which I have attached in a table for ease of reading.

Thank you for your important work in this area!

Reviewer #2: Comments for Authors

Abstract

1. The introduction is too long and quite repetitive. The last sentence on lines 39-42 could for, instance be summarized.

2. The topic is based on “Socio-economic, demographic, and behavioural determinants…”. I expect the results in the abstract on determinants to be clearly based on these as well.

3. The conclusion should reflect the finals based on the key variables in the topic. Line 61 for instance, “…determined by both individual and community factors,…” could be rather “… socio-economic, demographic, and behavioural factors” if those findings were made in the study. Otherwise, the title has to be changed to reflect what the authors are trying to communicate.

Keywords

4. I did not see any keywords.

Main manuscript

5. Aside from indicating in the topic and objectives that they sought to identify socioeconomic, demographic and behavioural determinants, I do not find the actual reporting of the authors’ findings as doing so. The authors should, therefore, ensure that there is consistency in what they seek to do and what they are actually communicating.

6. Why PCA? The authors should justify this somewhere on line 201.

7. I suggest the authors provide a table of the variables (not just for the indicators of empowerment [line 160]), the specific questions which represent them in the survey, and their coding. Additionally, a column should be provided in the table showing any recoding of the original variables used in the analyses.

8. I suggest the authors adopt the Plos One approved STROBE checklist in reporting their findings.

9. Line 215, the authors should clarify how the ORs and AORs were conducted. On what basis for instance, was a varible included in the AOR after conducting the COR. For instance, Age of first cohabitation was not significant in any of the CORs but was respectively included in the AORs.

10. Table 3. Instead of saying “OR (95% CI) adjusted”, I suggest the authors change this to AOR (95% CI). The current OR should also be changed to COR.

11. The strengths of the study should also be reported alongside the limitations. I suggest the authors create another sub-section after the discussion and name it “Strengths and limitations”.

12. A professional English Language editor should proof-read the work for typos and grammer.

6. PLOS authors have the option to publish the peer review history of their article (what does this mean?). If published, this will include your full peer review and any attached files.

Reviewer #1: No

Reviewer #2: No

---

## [Author Response · Author response to Decision Letter 0]

22 Dec 2020

Dear Editor

Dr. Eugene Kofuor Maafo Darteh,

PONE-D-20-25980

Socio-economic, demographic, and behavioural determinants of women’s empowerment in Mozambique. 

We are pleased to know that our manuscript was considered of interest to the PLOS ONE Journal and we thank you for the opportunity to revise and improve our manuscript based on the comments from the reviewers. Please find below our point-by-point response to the reviewers. 

We have revised and uploaded two versions of new manuscript – one with track-changes (page and line numbers provided from this version) and one without track-changes. Following the journal requirements, we also changed the formatting of the manuscript.

Hoping that the revised manuscript now meets all the requirements for publication, we thank you in advance for your consideration and look forward to hearing from you soon.

Sincerely,

Sofia Castro Lopes

Response to reviewers

REVIEWER #1

We thank the reviewer for insightful comments and suggestions which helped to improve the quality of the article. Please see our detailed responses to each specific comment and suggestion below.

Comments to the Author

1.Suggest defining empowerment and what is gap in measurement in Mozambique. Your aims listed lines 119-123 are very clear, something similar to this in the abstract would be useful (Page 2, line 37-42).

OUR RESPONSE#1: We changed the Introduction of the abstract to better reflect our aims. However, we did not include the definition of empowerment as the introduction was considered too long by reviewer#2. 

Abstract

Introduction (page 2, line 36)

“Empowerment is considered pivotal for how women access and use health care services and experience their sexual and reproductive rights. In Mozambique, women’s empowerment requires a better understanding and contextualization, including looking at factors that could drive empowerment in that context. This study aims to identify socioeconomic, demographic and behavioural determinants of different domains of women’s empowerment in Mozambique.“

2. Please list your results in the abstract by listing each domain and related contextual factors systematically to clarify (Page 2, line 53-59).

OUR RESPONSE#2: We changed the Results section in the Abstract to better describe and clarify the findings of the study, also taking into consideration the comments from reviewer 2 on this aspect. 

Abstract

Results (page 2, line 54)

“Three domains of women’s empowerment were identified, namely (1) Beliefs about violence against women, (2) Decision-making, and (3) Control over sexuality and safe sex. Region, rurality, the experience of intimate partner violence (IPV) and partner’s controlling behaviours were associated with Beliefs about violence against women, while Decision-making and Control over sexuality and safe sex were also associated with education, age and wealth. Employment, polygamous marriage and religion were positively associated with Decision-making, and access to media increased the odds of Control over sexuality and safe sex.” 

3. This is one definition, should make it clear that it is the one you are using, but not the only conceptualization of empowerment. Recommend you see Gram et al., 2018 

https://doi.org/10.1007/s11205-018-2012-2 Organising concepts of ‘women’s empowerment for measurement : a typology ( Page 4, line 91).

OUR RESPONSE#3: We are aware of the diversity of definitions that exist in the literature - an attempt to conceptualise such a complex term as empowerment. This has been a challenge across research developed in this area. Kabeer’s (1999) definition of empowerment is one of the most used and cited across the literature therefore we decided to provide that definition in the introduction of the paper. To include this important aspect highlighted by the reviewer we amended the text to reflect the existing diversity of definitions of empowerment in the literature.

Introduction (page 4, line 94)

“The definition of empowerment varies across the literature [2], but it can generally be described as the ability to exercise choice and free decision-making where this was previously denied [3]”

4. Would be helpful to include a bit more about participants and data collection for the survey: ie; is this urban and rural areas? Who collected the data, how are they trained, and how was data collected (ie paper, electronic etc). What is inclusion/exclusion criteria for women to be included in the survey. Not necessary to include all, but some would provide more depth to those not familiar with DHS data (Page 4, line 138-144).

OUR RESPONSE#4: We added more detailed information about the DHS 2015 conducted in Mozambique. 

Methods (page 6, line 142)

“The DHS followed a rigorous population sampling process to ensure national, regional, urban, and rural representativeness. The survey was piloted in non-selected areas of the country, and changes made for improving clarity and adequacy of the questions. Interviewers received theoretical and practical training for the field work. A total of 25 teams were organised and distributed across the country. These included a supervisor, interviewers, and a person responsible to capture the data electronically. 

Data was collected from all women and men, aged 15 to 59, residents or visitors that spent the night prior to the interview in one of the selected households. From the 7368 selected households, a total of 7129 were included in the survey, and 7749 women and 5283 men were interviewed [21]. Data were initially collected in paper form, and immediately entered electronically in a database.” 

5. Please cite the sentence of your empowerment definition (Page 4, lines 150-152)

OUR RESPONSE#5: The definition was developed and proposed by us, the authors. As explained in the same paragraph of the methods section, this definition was based on Kabeer’s definition and in the Capability model adapted to health from Amartya Sen’s work. We changed the text to clarify this point.

Methods (page 7, line 165)

“This definition is based on Kabeer’s conceptualization of empowerment [3] and it incorporates the capability approach as a well-being measurement initially developed by Amartya Sen [24] and later adapted to health and empowerment studies [22].”

6. Please describe briefly rationale for combining regions in the way that was done (Page 4, lines 173-174)

OUR RESPONSE#6: We amended the text as per the reviewer’s indication. 

Methods (page 8, line 187)

“The 11 provinces were used in the analysis and combined in three regions following the official aggregation of provinces by the Mozambique Government [26]: South – Maputo city, Maputo Province, Inhambane and Gaza; Centre – Sofala, Manica, Tete and Zambezia; and, North – Niassa, Cabo Delgado and Nampula.”

7. Methods would be strengthened with rationale for use of orthogonal rotation- ie were other rotations attempted? Lines 200-206

OUR RESPONSE#7: The decision of applying an orthogonal rotation was based on the uncorrelation of the retained components. We changed the text to better reflect the decisions and steps taken.

Data analysis (page 9, line 222)

“An orthogonal varimax rotation was applied after confirming no correlation between the retained components, an essential criterium for this type of rotation [31–33].”

8. Table 1. Title should stand alone: can you add something else to make it clear what is being displayed? 

OUR RESPONSE#8: We changed the title of Table 1 to make it clearer and more specific. 

Results (page 10, line 256) 

“Table 1: Socioeconomic, demographic and behavioural characteristics of women included in the study.” 

9. Table 3 is very comprehensive---is there a way to simplify these results or break them up into other sections? Consider removing Table 1 or adding it as a supplement to make more space to share the results in Table 3 broken into separate tables. 

OUR RESPONSE#9: Although we understand that Table 3 is quite comprehensive, we believe that these results should be presented in one single table. We think this way of presenting the results, allows a better overview and comparison across the different domains of empowerment. Therefore, we decided to keep the Table 3 as well as 1 and 2 in the main body of the article. 

10. Consider adding a sentence or two of literature that supports this finding. Line 290

OUR RESPONSE#10: We added a sentence referring to evidence from other studies, to support our findings. 

Discussion (page 18, line 334)

“The importance of community factors for impeding or facilitating women’s empowerment and its relationship with violence against women has been described in other contexts [37]. Additional to community and/or contextual determinants, individual characteristics of women like education, age or wealth, also seem to play a role in determining empowerment in the domain Decision-making and Control over sexuality and safe sex in Mozambique, similar to findings from other studies [35,36]. Our findings could be explained by the fact that women’s individual beliefs are rooted in socio-cultural norms and traditional practices embedded in patriarchal systems, learned and maintained by the community where women live [38,39] and oftentimes perpetuated by women themselves [37,39,40]. Despite the matrilineal societal organization of the north region, Mozambique is a patriarchal society with rigid gender norms that retain men in power positions [41,42]. While education, wealth, age, and employment are assets or resources that women use in the process of decision-making and choice [36], therefore playing an important role in the other two domains: Decision-making and Control over sexuality and safe sex.”

11. Consider splitting this into two sentences: 1 indicating other research that indicates the individual characteristics that impact empowerment, and 2 how this played out in this sample in Mozambique. Line 293-295

OUR RESPONSE#11: Although we did not split the sentence into two, we changed the text to better reflect the reviewer’s suggestion.

Discussion (page 18, line 337)

“Additional to community and/or contextual determinants, individual characteristics of women like education, age or wealth, also seem to play a role in determining empowerment in the domain Decision-making and Control over sexuality and safe sex in Mozambique, similar to findings from other studies [35,36].”

12. Please provide citation for this. Line 307-309

OUR RESPONSE#12: The following citation was included: 

“Ahinkorah BO, Dickson KS, Seidu A-A. Women decision-making capacity and intimate partner violence among women in sub-Saharan Africa. Arch Public Health. 2018;76: 5. doi:10.1186/s13690-018-0253-9”

13. Please look into work by Dr. Sidney Schuler indicating what happens longitudinally when gender norms are transgressed and consider including this. Line 310-312

OUR RESPONSE#13: We changed the text and included a reference to Dr. Sidney Schuler work to better discuss the positive association of IPV and empowerment. 

Discussion (page 19, line 355)

“Furthermore, available evidence suggests that when women enter the pathway of empowerment they may challenge gender norms and gender power relations, which might initially expose them to a greater risk of experiencing violence and controlling behaviours perpetrated by partners, referred to as violence backlash [37,38,43,46]. However, there is evidence suggesting that empowerment can become protective against IPV throughout time, where the empowered women are less likely to be perceived as transgressing the gender norms [47].”

REVIEWER #2

We thank the reviewer for the relevant, insightful, and useful comments which have contributed substantially to improve the quality and clarity of the manuscript. 

Comments to the Author

Abstract

1. The introduction is too long and quite repetitive. The last sentence on lines 39-42 could for, instance be summarized.

OUR RESPONSE#1: We amended and summarized the text accordingly.

Abstract 

Introduction (page 2, line 40)

“This study aims to identify socioeconomic, demographic and behavioural determinants of different domains of women’s empowerment in Mozambique.”

2. The topic is based on “Socio-economic, demographic, and behavioural determinants…”. I expect the results in the abstract on determinants to be clearly based on these as well.

OUR RESPONSE#2: We have changed the results and conclusion of the abstract to clearly reflect the topic and aim of the study. 

Results (page 2, line 55)

“(…) Region, rurality, the experience of intimate partner violence (IPV) and partner’s controlling behaviours were associated with Beliefs about violence against women, while Decision-making and Control over sexuality and safe sex were also associated with education, age and wealth. Employment, polygamous marriage and religion were positively associated with Decision-making, and access to media increased the odds of Control over sexuality and safe sex.” 

3. The conclusion should reflect the finals based on the key variables in the topic. Line 61 for instance, “…determined by both individual and community factors,…” could be rather “… socio-economic, demographic, and behavioural factors” if those findings were made in the study. Otherwise, the title has to be changed to reflect what the authors are trying to communicate.

OUR RESPONSE#3: Aligned with the previous comment, we have changed the conclusion of the abstract to better reflect and answer the aim of the study.

Conclusion (page 3, line 64)

“Women’s empowerment seems to be determined by different socioeconomic, demographic, and behavioural factors, and this seems to be closely related to different domains of empowerment identified. This finding affirms the multi-dimensionality of empowerment as well as the importance of considering the context- and community-specific characteristics.” 

Keywords

4. I did not see any keywords.

OUR RESPONSE#4: The authors apologise for overlooking the keywords. During the submission the corresponding author included keywords in the submission platform however by mistake did not include them in the manuscript.

Keywords (page 3, line 69) 

“Empowerment; women; social determinants; Mozambique” 

Main manuscript

5. Aside from indicating in the topic and objectives that they sought to identify socioeconomic, demographic and behavioural determinants, I do not find the actual reporting of the authors’ findings as doing so. The authors should, therefore, ensure that there is consistency in what they seek to do and what they are actually communicating.

OUR RESPONSE#5: We have changed the results section to increase consistency and clarity throughout the manuscript. We also updated the conclusion section. This was informed by the STROBE checklist as suggested in point 8.

Results (page 13, line 280)

“Table 3 shows the crude and adjusted OR for the association between the socio-economic, demographic, and behavioural characteristics and the different domains of empowerment. After adjusting for woman’s education, we observed that age, education, current employment, age of first cohabitation, polygamous marriage and the wealth index were not associated with the domain Beliefs about violence against women. However, experiencing at least one type of controlling behaviour, being exposed to IPV, having access to media, and living in the South region of Mozambique was significant and negatively associated with being empowered in this domain, which seems to indicate that these factors are determinants of lower levels of empowerment for Beliefs about violence against women. Rurality had significant and positive impact on this domain.

After adjusting for education, Decision-making domain of empowerment was significantly and positively associated with women of older age, more educated, currently working, living in South or Centre regions and with increased levels of wealth (Table 3). IPV and controlling behaviours from the partners, were also statistically and positively associated with higher decision-making power. No associations were found between this domain of empowerment and age of first cohabitation, access to media and rural vs. urban residency. 

Current employment, age at first cohabitation, polygamous marriage and religion were not associated with Control over Sexuality and safe sex after adjusting for women’s education. However, having some education, living in the South region, being among the richest wealth quintile, having access to media at least once a week as well as experiencing IPV or partner’s controlling behaviour had a significant and positive impact in women’s empowerment level for this domain, after adjusting for education (Table 3). Being 40 to 49 years old and living in a rural area were significantly and negatively associated with women’s control over their sexuality.”

Conclusion (page 21, line 414)

“In Mozambique, women’s empowerment seems to be determined by socioeconomic, demographic, and behavioural factors, and this seems to be closely related to the different domains of empowerment identified.”

6. Why PCA? The authors should justify this somewhere on line 201.

OUR RESPONSE#6: We added a brief explanation about what a PCA is and the justification of its use in this study.

Data analysis (page 9, line 215)

“Then, a principal component analysis (PCA) was carried out. PCA is a technique to transform a data set with a large number of indicators into a smaller data set of uncorrelated indicators, while capturing as much as possible of the variation of the original data set [29]. This procedure allows assessment of clustering patterns of empowerment indicators and the contribution (weight) for each component. PCA has been applied in studies on women’s empowerment to avoid ad hoc estimation of summary scores in which each indicator has an equal contribution [8,23,30] From the scree plot of the PCA results, the significant components (eigenvalue above 1) were retained.”

7. I suggest the authors provide a table of the variables (not just for the indicators of empowerment [line 160]), the specific questions which represent them in the survey, and their coding. Additionally, a column should be provided in the table showing any recoding of the original variables used in the analyses.

OUR RESPONSE#7: We included in the original submission supporting information which contained a table (S1 Table) with a detailed description of the selected empowerment indicators and the coding applied for the purpose of this study. We have taken into consideration the reviewer’s suggestion and we expanded the table to include the codes used in the survey as well as the recoding of each question for this study. The S1 Table was resubmitted with track-changes. 

8. I suggest the authors adopt the Plos One approved STROBE checklist in reporting their findings.

OUR RESPONSE#8: We have used the STROBE checklist to review our article, focussing particularly on the findings section (please see response to comment 5). The STROBE checklist can be found attached.

9. Line 215, the authors should clarify how the ORs and AORs were conducted. On what basis for instance, was a variable included in the AOR after conducting the COR. For instance, Age of first cohabitation was not significant in any of the CORs but was respectively included in the AORs.

OUR RESPONSE#9: The inclusion of “Age of first cohabitation” in the final model was decided a priori, based on theoretical reasoning. This is a relevant variable for the empowerment of women in developing countries and therefore we decided to include it. We added an explanation to the methods section, following the indications from the STROBE checklist as well. 

Data analysis (page 10, line 236)

“(…) The final models were adjusted for women’s education, as research has shown that education is strongly associated with both empowerment and the other socio-economic, demographic, and behavioural characteristics included in the study. By adjusting for education, we aimed to assess if the associations found between the selected characteristics and empowerment were independent of the education level of women. The inclusion of the different characteristics in the final models were informed by both theoretical and/or statistical justification (significance level set at 0.05).”

10. Table 3. Instead of saying “OR (95% CI) adjusted”, I suggest the authors change this to AOR (95% CI). The current OR should also be changed to COR.

OUR RESPONSE#10: We thank the reviewer for pointing this out. We changed the column headings following the reviewer’s suggestion. We also updated the text in the finding’s text to reflect this change. 

11. The strengths of the study should also be reported alongside the limitations. I suggest the authors create another sub-section after the discussion and name it “Strengths and limitations”.

OUR RESPONSE#11: We added the subheading “Strengths and limitations” after the discussion as per the reviewer’s suggestion. We updated the text accordingly. 

Strengths and limitations (page 20, line 392)

“The study’s strength is related to the use of a large sample of women of reproductive age, from a population-based survey, which allowed generalizability of the findings for Mozambique. Nevertheless, the study has some limitations that should be considered.”

12. A professional English Language editor should proof-read the work for typos and grammar.

OUR RESPONSE#12: We ensured that the final manuscript was proofread for typos and grammar.

---

## [Decision Letter · Decision Letter 1]

14 May 2021

Socio-economic, demographic, and behavioural determinants of women’s empowerment in Mozambique.

PONE-D-20-25980R1

Dear Dr. Sofia Castro Lopes,

We’re pleased to inform you that your manuscript has been judged scientifically suitable for publication and will be formally accepted for publication once it meets all outstanding technical requirements.

Kind regards,

Eugene Kofuor Maafo Darteh, Ph.D.

Academic Editor

PLOS ONE

Additional Editor Comments (optional):

Reviewers' comments:

Reviewer's Responses to Questions

**Comments to the Author**

1. If the authors have adequately addressed your comments raised in a previous round of review and you feel that this manuscript is now acceptable for publication, you may indicate that here to bypass the “Comments to the Author” section, enter your conflict of interest statement in the “Confidential to Editor” section, and submit your "Accept" recommendation.

Reviewer #1: All comments have been addressed

2. Is the manuscript technically sound, and do the data support the conclusions?

Reviewer #1: Yes

3. Has the statistical analysis been performed appropriately and rigorously? 

Reviewer #1: Yes

4. Have the authors made all data underlying the findings in their manuscript fully available?

Reviewer #1: Yes

5. Is the manuscript presented in an intelligible fashion and written in standard English?

Reviewer #1: Yes

6. Review Comments to the Author

Reviewer #1: (No Response)

7. PLOS authors have the option to publish the peer review history of their article (what does this mean?). If published, this will include your full peer review and any attached files.

Reviewer #1: No

---

## [Editor Report · Acceptance letter]

19 May 2021

PONE-D-20-25980R1 

Socio-economic, demographic, and behavioural determinants of women’s empowerment in Mozambique 

Dear Dr. Castro Lopes:

I'm pleased to inform you that your manuscript has been deemed suitable for publication in PLOS ONE. Congratulations! Your manuscript is now with our production department. 

Kind regards, 

on behalf of

Dr. Eugene Kofuor Maafo Darteh 

Academic Editor

PLOS ONE